# The Unseen Side of Feline Hypertrophic Cardiomyopathy: Diagnostic and Prognostic Utility of Electrocardiography and Holter Monitoring

**DOI:** 10.3390/ani14152165

**Published:** 2024-07-25

**Authors:** Alexandra Cofaru, Raluca Murariu, Teodora Popa, Cosmin Petru Peștean, Iuliu Călin Scurtu

**Affiliations:** 1Department of Small Animal Internal Medicine, University of Agricultural Sciences and Veterinary Medicine, 400372 Cluj-Napoca, Romania; alexandra.cofaru@usamvcluj.ro (A.C.); dorrapopa@gmail.com (T.P.); iuliu.scurtu@usamvcluj.ro (I.C.S.); 2Department of Surgical Techniques and Propaedeutics, University of Agricultural Sciences and Veterinary Medicine, 400372 Cluj-Napoca, Romania; cosmin.pestean@usamvcluj.ro

**Keywords:** hypertrophic cardiomyopathy, arrythmias, electrocardiography, Holter monitoring, diagnostic, prognostic

## Abstract

**Simple Summary:**

Hypertrophic cardiomyopathy is the most common acquired heart disease of cats. Arrhythmias are frequently observed in cats with hypertrophic cardiomyopathy, contributing to the complexity of the condition. Understanding rhythm disturbances associated with feline hypertrophic cardiomyopathy is crucial, as they could contribute to diagnosing the condition, risk stratification, and therapeutic management. By analyzing the current level of knowledge regarding arrythmias associated with feline hypertrophic cardiomyopathy, this study aimed to assess the utility of available diagnostic methods. Even though progress has been made in understanding arrythmias linked to hypertrophic cardiomyopathy, further research is needed to better assess their diagnostic and prognostic value, as well as the potential benefits of anti-arrhythmic therapy for affected cats.

**Abstract:**

Hypertrophic cardiomyopathy (HCM) is a common heart disease in cats, characterized by regional or diffuse hypertrophy of the left ventricular walls, with an uncertain etiology and heterogenous natural history. Several types of rhythm disturbances are often associated with the disease. This study conducts a comprehensive review of the current literature, in order to evaluate the diagnostic and prognostic effectiveness of electrocardiography and Holter monitoring in the management of feline hypertrophic cardiomyopathy. The main subjects of discussion will include general information about HCM and its connection to arrhythmias. We will explore the rhythm disturbances documented in the current literature on Holter monitoring, as well as the techniques used for Holter monitoring. Additionally, the review will cover classical electrocardiography (ECG) and its diagnostic utility. Prognostic indicators and anti-arrhythmic therapy will also be discussed in detail. The findings highlight the importance of understanding arrhythmias in feline HCM for accurate diagnosis, risk assessment, and therapeutic intervention. ECG and Holter monitoring may offer valuable insights into managing feline HCM.

## 1. Introduction

### 1.1. Overview of Hypertrophic Cardiomyopathy in Cats

Feline hypertrophic cardiomyopathy (HCM) represents the most common acquired cardiac disease affecting cats, with a prevalence of approximately 15% in the general cat population, but clear prevalence data for pedigree cats is lacking [1]. It is characterized by variable thickening of the left ventricular wall, in the absence of any concurrent disease capable of producing this hypertrophy, such as hyperthyroidism or systemic hypertension [2,3]. Other causes that can produce left ventricular hypertrophy and mimic HCM are neoplastic infiltration, reduced preload (dehydration, diuretic administration), transient myocardial thickening, acromegaly [1], and (sub)aortic stenosis [4]. This disease exhibits great heterogeneity, in both progression and clinical expression [3]. While most cats remain asymptomatic despite the hypertrophy of the left ventricle, others can develop complications, most notably, congestive heart failure (CHF), arterial thromboembolism (ATE), or sudden cardiac death (SCD) [5]. The factors that influence the transition from an occult state to a symptomatic one are not well known. However, genetic factors are implied, as well as systemic and metabolic disorders and concurrent disease, anesthesia, and fluid administration [6].

Sarcomeric gene mutations associated with HCM have been identified in some cat breeds, including Maine Coon, Ragdoll, and Sphynx [7] The cause for most Maine Coon and Ragdoll cats that develop HCM is a mutation in the myosin-binding protein C genes MyBPC3-A31P and MyBPC3-R818W (originally known as R820W), respectively [1,7,8]. Both mutations have been demonstrated to be pathogenic in Maine Coon and Ragdoll cats [8]. Maine Coon cats primarily develop clinically relevant HCM when they are homozygous for the A31P mutation [1], as homozygosity is linked with high HCM penetrance and a high risk of cardiac death [9]. Those that are heterozygous for this mutation might experience only discrete systolic and diastolic dysfunction, potentially remaining subclinical [7]. However, heterozygous cats can also develop clinical signs and severe forms of the disease [9]. Ragdoll cats that are homozygous for the mutation are also predisposed to develop severe forms of HCM and have a shorter survival time [10]. A mutation to the ALMS1 gene in Sphynx cats has been positively associated with the presence of HCM [11]. Other breeds that are predisposed to developing HCM are British Shorthair, Persian, Bengal, Norwegian Forest, and Birman cats, but most cats with HCM are of a domestic breed [1]. The implication of non-genetic factors is unknown [1].

Macroscopic changes specific for HCM are concentric hypertrophy of the interventricular septum, left ventricular free wall and papillary muscles, and left atrial enlargement. The hypertrophy can be symmetrical (generalized) or asymmetrical (focal) [3]. Histologically, HCM is characterized by myofiber hypertrophy and disarray, variable interstitial fibrosis, changes in the coronary arteriolar microvasculature, such as intramural coronary arteries atherosclerosis, and even inflammatory infiltrates [3,12].

The gold standard diagnostic test for HCM is echocardiography [6]. However, differentiating mild disease from a normal phenotype poses a challenge, with a narrow margin for error [13]. Additional diagnostic tests, such as genetic testing, cardiac biomarkers, radiography, and electrocardiography (ECG), can aid in disease staging, detecting significant comorbidities, and determining the prognosis [1]. Other tests are recommended to detect potential underlying causes of hypertrophy, such as evaluating the serum thyroxine concentration or measuring blood pressure [1]. Cardiac magnetic resonance imaging plays a key role in the diagnosis of human HCM, being used in the quantification of left ventricular mass and function. However, it is not commonly used in feline patients [14,15]. In recent times, more advanced methods for postmortem diagnosis have emerged. Micro-CT can now be used to determine myocardial mass and to characterize myocardial disarray and fibrosis [16].

### 1.2. Importance of Holter Monitoring in HCM Evaluation

While echocardiographic changes that appear in HCM are well defined, rhythm disturbances are unpredictable, and most of the time, inconstant. Their prognostic significance and clinical relevance are also quite unknown [6,17].

Arrhythmias are classified in three main categories, based on the mechanism responsible for their formation: abnormal impulse formation, conduction disturbances, or a combination of both. The first class includes arrhythmias caused by abnormal automaticity or triggered activity and the second refers to conduction blocks [18].

The association between left ventricular hypertrophy (LVH) and rhythm disturbances is grounded in a logical morphological change, with the structural anomalies that appear in HCM offering a foundation for cardiac conduction and excitability abnormalities, specifically those originating from the ventricles [19]. The primary cause of electrical instability leading to arrhythmias in HCM is the disorganized structure of the left ventricular myocardium, meaning a chaotic pattern of myocyte arrangement. Additionally, myocyte death and replacement by fibrous tissue caused by changes in the microvasculature and transient myocardial ischemia episodes contributes to arrythmia formation [20].

Arrythmias associated with HCM are usually ventricular premature complexes (VPCs) with or without complexity [12,21], but atrial premature complexes (APCs) may also be observed [19,21,22]. These types of arrhythmias can also be linked with other types of cardiomyopathies [23]. Other supraventricular arrythmias that may be encountered are atrial fibrillation and supraventricular tachycardia [4,24]. Atrial fibrillation is demonstrated to occur more commonly in cats with severe atrial enlargement caused by cardiac disease and is associated with a negative outcome [25]. In people, atrial fibrillation could be linked to an increased risk of sudden death [26].

In humans, ECG and Holter monitoring are used broadly in the assessment and risk stratification of patients with HCM [27], with non-sustained ventricular tachycardia being established as a risk marker for sudden cardiac death [28]. This could also apply in the feline disease, as HCM in cats exhibits remarkable resemblances at the molecular, histopathological, clinical, and genetic levels to human HCM [14,29]. While some studies investigate the correlation between arrythmias and the risk of sudden cardiac death in cats [30], Holter monitoring is not routinely used in the diagnosis and management of feline HCM.

The aim of this study is to review the diagnostic and prognostic effectiveness of electrocardiography and Holter monitoring in the management of feline hypertrophic cardiomyopathy through an extensive analysis of the current literature, focusing on what types of arrythmias have been identified and their significance.

## 2. Ventricular and Supraventricular Arrythmias on Holter Monitoring

Over the past 25 years, several clinical studies have documented the use of 24 h Holter ambulatory ECG in healthy cats and patients with HCM or other forms of cardiomyopathy (Table 1). A variety of ventricular or supraventricular arrhythmias have been shown to occur in all categories of cats, with diverse frequency and complexity, highlighting the fact that rhythm disturbances are an important part of the pathogenesis of feline cardiomyopathies and their comprehensive characterization using Holter monitoring and ECG could contribute to the management of these feline patients. The diversity of the results and even opposite outcomes between different studies emphasize the complexity of the disease and the need for filling the current knowledge gap on the subject.

While some of these studies show a significant difference in the presence and complexity of arrythmias between healthy and affected cats [12,21], others show that the overall frequency and complexity of rhythm disturbances in these two categories are similar [22,31].

One study that compares clinically healthy cats, cats with compensated HCM (cHCM), and cats with decompensated HCM (dHCM) shows that VPCs were present in 8/10 control cats, 15/16 cats from the second group, and 15/15 cats from the last group. Ventricular ectopy was more common and complex in cHCM and dHCM cats compared to healthy controls and the difference was statistically significant [12]. While none of the cats from the control group showed episodes of ventricular tachycardia, 8/16 cats with cHCM and 6/15 cats with dHCM presented with this type of arrythmia. However, this study did not underline a statistical difference between the two HCM groups [12].

Jackson also demonstrated that cats with HCM showed more frequent and complex supraventricular and ventricular arrythmias than healthy cats, in a study conducted on 17 feline patients with HCM. Ventricular arrythmias were identified in all affected cats and 82% of them had complex arrythmias. Ventricular ectopic beats were present in 14/15 control cats, with only 20% exhibiting complexity [21].

However, a study conducted by Hanås demonstrated that the number of VPCs and APCs was not different between HCM-affected cats and healthy controls. Only asymptomatic patients with HCM were included in this study. Out of 15 cats with HCM, 12 showed ventricular ectopy, ranging from 0 to 1745 complexes. Bigeminy and couplets were also observed [31]. In a previous study of the author, 18/23 healthy cats showed between 0 and 1745 ventricular ectopic events, with complexity being identified as well [32].

In human HCM, an increased risk of sudden cardiac death is linked to ventricular arrythmias during exercise. Unfortunately, exercise testing is not available in cats. One study tried to implement a pharmacological cardiac stress test using oral administration of terbutaline, but this test failed to demonstrate an effect on the frequency and complexity of ventricular or supraventricular arrythmias. One explanation for this result was the fact that the authors included only cats with mild disease, meaning only feline patients without clinical signs or evidence of congestive heart failure. This could be the reason for the fact that there was no difference regarding the frequency and complexity of arrythmias between healthy controls and cats with HCM. Ventricular ectopic beats were observed in all 16 cats with HCM and 6/7 healthy controls, with a range of 1–202, and 0–151, respectively. Episodes of ventricular tachycardia appeared in both groups. A significant association was identified between the genotype for an HCM-causative mutation and the frequency of supraventricular and ventricular ectopy [22].

One study conducted on cats with other forms of cardiomyopathies, such as restrictive cardiomyopathy (RCM), arrhythmogenic right ventricular cardiomyopathy (ARVC), and non-specific cardiomyopathy, shows that potentially significant ventricular arrhythmias are prevalent in cats with non-hypertrophic cardiomyopathy (NHCM) and Holter monitoring could be a valuable diagnostic tool. All 13 cats included in this study showed between 338 and 8305 ventricular complexes. Complex arrythmias were observed, such as bigeminy events, couplets, and ventricular tachycardia [23].

Various results have been described regarding supraventricular arrhythmias. In a study by Bartoszuk, supraventricular ectopic beats appeared only in cats with HCM, with two cats with cHCM and two cats with dHCM showing isolated APCs and episodes of supraventricular tachycardia [12]. Jackson demonstrated that supraventricular ectopic beats were present in 15/17 cats with HCM and 9/15 healthy control cats [21]. In two studies conducted by Hanås, very few supraventricular ectopic beats were identified in both healthy cats and cats with HCM. Specifically, three cats with HCM had two–four complexes, and one healthy cat had one complex [31,32].

Some of these studies were conducted in the home environment [21,31] and others in the hospital environment [12,22,23]. To the authors’ knowledge, it is unknown whether the stress of hospitalization has any influence on the frequency and complexity of arrythmias. It has been shown that healthy cats have a lower heart rate at home compared to in hospital environments, so hospitalization and restraint are stress factors that significantly increase the heart rate [33]. This raises the question as to whether stress and a high sympathetic tone influence Holter monitoring findings, as the presence of the device can itself be a stress factor. It is believed that ventricular arrhythmias could be triggered by an elevated sympathetic tone [12]. One study conducted on healthy cats that used a Patch Holter device demonstrated that the occurrence of arrythmias was no different when using this kind of monitoring for multiple days, nor when the results were compared to a conventional Holter device [34].

Other types of arrythmias have been identified on Holter monitoring and consisted of abnormal impulse formation (atrial fibrillation) or conduction (complete atrio-ventricular block, atrial standstill) [12,23]. These rhythm disturbances were observed in cats with HCM or other forms of cardiomyopathies (restrictive cardiomyopathy, arrhythmogenic right ventricular cardiomyopathy, non-specific cardiomyopathy) [12,23].

Ventricular and supraventricular arrythmias also occur in clinically healthy cats, free of any form of cardiomyopathy [32,35], more commonly in older cats [36]. In a study conducted on 20 healthy cats, 6 showed between 1 and 18 isolated VPCs and 2 showed 3–4 isolated APCs [35]. A recent study demonstrated that the arrythmias were more likely to be present in older cats and males [37].

## 3. Holter Monitoring Technique

Different electrode placement methods have been used for continuous ECG monitoring. These methods are summarized in Table 2. To our knowledge, a standard method for electrode placement in Holter monitoring is not yet established for cats. A high T wave and small-QRS-complex characteristic for cats offer room for error in the total number of QRS complexes [38], so adequate placement of the electrodes could improve the quality of the examination.

The most used method employed five electrodes, either two on the right and three on the left side of the thorax, or three on the right and two on the left side [22,23,36]. Bartoszuk used three electrodes, with two placed on the left side of the thorax, and one on the right [12]. Hanås used seven electrodes that provided a three-lead ECG [32].

## 4. Electrocardiography and Its Diagnostic Utility in HCM

In human HCM, the electrocardiogram plays an important role in the early identification of disease, differential diagnosis, and risk stratification, as certain patterns of a 12-lead ECG are strongly indicative and may be the sole indication of disease in its early stages [39]. This statement has not been supported by studies on feline HCM, as electrocardiography has turned out to have a rather insensitive diagnostic value [13].

For example, the presence of arrhythmias on electrocardiography had a sensitivity of 31% and specificity of 100% for identification of left ventricular hypertrophy, when the ECG was recorded for 2 min. Several types of arrythmias were associated with left ventricular hypertrophy, such as supraventricular and ventricular ectopic beats, accelerated idioventricular rhythm, and ventricular tachycardia, but also second- and third-degree atrio-ventricular block (AVB), and sinus bradycardia [19]. Although electrocardiography has a low sensitivity in identifying left ventricular hypertrophy, its specificity should not be ignored, as it has been demonstrated that out of 106 cats with ventricular arrythmias on ECG, 102 presented echocardiographic abnormalities consistent with different types of cardiomyopathies [40]. Regarding the prognostic utility of surface electrocardiography, the only variables that were statistically different between healthy cats and those with left ventricular hypertrophy were the duration of QT interval and the QT interval corrected for heart rate (QTc), which could be linked to survival time [19].

In a study conducted by Ferasin, out of 61 cats diagnosed with HCM, 26 cats (42.6%) did not exhibit any notable abnormalities on the ECG. The only arrhythmias identified were singular ventricular ectopic beats, and they were present in only a few feline patients. The most notable rhythm disturbances identified were left anterior fascicular block and changes correlated with LV enlargement (QRS complexes with a duration > 0.04 s and R waves with an amplitude > 0.9 mV) [2].

In human cardiomyopathies, T wave inversion on a 12-lead ECG (defined as a negative T wave > 1 mm in two or more contiguous leads besides aVR, III, and V1) could be an early indicator of underlying occult structural heart disease and may suggest possible adverse outcomes such as sudden death [41]. An ECG pattern consisted of ST-segment depression and T wave inversion, called a strain pattern, has also been associated with developing LV hypertrophy and cardiovascular events [42]. To the authors’ knowledge, these parameters have not been studied for feline cardiomyopathies.

## 5. Heart Rate and Heart Rate Variability

In people with HCM, the average heart rate turned out to have prognostic value, being significantly lower in patients that later developed cardiovascular events, even though it did not differ between HCM patients and the healthy control group [28].

In healthy cats, the average heart rate varies with age and gender [31]. In female cats, minimum, maximum, and average heart rates have been demonstrated to be higher than in males [36]. Mean heart rates were the lowest during the night [36]. Furthermore, heart rate appears to be higher when recorded in hospital conditions, compared to home environments [33].

When analyzing mean heart rates in cats affected by HCM, results are contradictory. Three studies that used Holter monitoring demonstrated that there is no difference in mean heart rate between HCM-affected cats and healthy cats [21,22,31]. In two of these studies, Holter monitoring was conducted at home [21,31], and in the third one, in the hospital environment [22]. However, a different study conducted in a hospital environment showed that cats with cHCM and dHCM had higher mean heart rates than healthy controls, but there was no difference between the two HCM groups [12]. Heart rate seemed to be higher during the first hours of wearing the device [12,21,31]. To the author’s knowledge, a link between average heart rate and prognosis has not been established.

Heart rate variability (HRV) refers to the variation in the time intervals between consecutive heartbeats. HRV can be used to assess autonomic function by evaluating sympathovagal balance and HRV profiles seem to vary depending on environmental factors that can produce stress, for example, between the home and hospital environments [33]. When recorded for 24 h, it can be used for stratification of cardiac risk in humans. One of the measurements used for assessing heart rate variability is the standard deviation of the inter-beat interval of normal sinus beats (SDNN) [43]. Human patients at high risk of death due to CHF can be identified by a reduction in the SDNN [44].

Very little is known about HRV in cats. In healthy cats, HRV varies depending on environmental conditions (in the hospital environment vs. the home environment) [33]. To our knowledge, heart rate variability in cats with HCM has only been evaluated in a single study, although it could be a useful marker for risk stratification. Cats with the HCM phenotype showed a greater HRV and a higher SDNN median value, compared to control cats. The difference in HRV profiles was statistically different in this study (*p* = 0.0001) [22]. These findings are contrary to what was anticipated because of information from human medicine and further studies are necessary to evaluate HRV as a prognostic marker for cats.

## 6. Arrythmias and Their Prognostic Significance in Feline HCM

Although several echocardiographic factors have been associated with survival time, such as lower left atrial fractional shortening (LA-FS%) [45], it is unclear whether arrhythmias are negatively associated with survival or not. To the authors’ knowledge, there is no study that demonstrates a link between fibrosis, ventricular arrhythmias, and the risk of SCD in cats, even though cats who experience SCD have been found to exhibit a greater prevalence of interstitial fibrosis, subendocardial fibrosis, and intramural arteriolosclerosis upon necropsy [46].

The presence of arrythmias has been demonstrated to be an univariable predictor of elevated risk of cardiac mortality in cats, as it can be linked to advanced disease states [17]. Besides the association with an increased risk of sudden death, arrythmias have also been linked with death by CHF [30]. However, details such as the types of arrythmias, or their incidence or complexity are not known.

One study that documented the presence and nature of arrythmias in cats with HCM stated that even though affected cats had a higher incidence of arrythmias, with a higher grade of complexity, the number and complexity of ventricular arrhythmias could not be established as a risk factor for sudden cardiac death or other causes of cardiac death in cats with HCM [12].

## 7. The Use of Anti-Arrhythmic Therapy

While arrhythmias are frequently observed in cats with subclinical or clinical HCM, neither the necessity for therapy nor a standardized therapeutic approach has been established. Currently, there is insufficient evidence to support the efficacy of any specific therapy in reducing morbidity or extending the lifespan of cats affected by occult HCM [6].

Atenolol, at a dose of 6.25–12.5 mg PO q12h, reduced the total number of ventricular complexes (from a geometric mean of 123/24 h to 22/24 h) and VPCs (from a geometric mean of 61/24 h to 15/24 h) in a study involving 17 cats with subclinical HCM. The complexity of ventricular ectopy was also decreased [47]. However, the link between ventricular arrythmias and sudden death in cats with HCM is not well defined, so it is unclear if therapeutic reduction in the number of ventricular ectopies could improve survival time. In a study that included 63 cats with HCM and 31 healthy control cats, atenolol had no effect on the 5-year survival time of cats that had subclinical HCM [48]. Beta-blockers should never be used when CHF is present, as it can aggravate decompensation [4]. One study tried to assess the impact of oral atenolol therapy on the quality of life of cats with subclinical HCM, but the results did not indicate a therapeutic advantage of atenolol [49].

Some authors recommend treatment for atrial fibrillation if the ventricular rate is higher than 250 bpm. Treatment could include diltiazem (1–2.5 mg/kg PO q8h or 7.5–15 mg/cat PO q8h) and atenolol (6.25–12.5 mg/cat PO q12h). Digoxin is not recommended in cats due to its toxicity [4].

## 8. Discussion

Arrythmias have been known to be a frequent and clinically relevant complication of HCM and other forms of cardiomyopathies in cats. However, differences in study outcomes summarized in this review, concerning arrhythmia prevalence and complexity in HCM, highlight the diverse nature of this disease and emphasize the necessity for additional studies to determine more consistent patterns. While the findings of some studies indicate a considerable contrast in the prevalence and complexity of arrhythmias between healthy cats and those with HCM [12,21], others suggest that results are similar in cats with HCM and healthy cats [22,31].

Holter monitoring is a useful diagnostic test for the characterization of arrythmias, with the occurrence of significant arrhythmias being higher than previously assessed through routine electrocardiography [50]. It is widely used in the early diagnosis of cardiomyopathies in dogs, such as dilated cardiomyopathy (DCM) in Dobermann Pinchers and ARVC in Boxers [51,52]. However, Holter monitoring is not currently used as a routine test for HCM in cats, and its diagnostic and prognostic value remains unclear but not of minor importance. Although the presence of arrythmias on classic electrocardiography has a high specificity of identifying left ventricular hypertrophy [19], ECG has proven to be relatively insensitive as a diagnostic tool [2,13].

While arrhythmias are frequently observed in cats with HCM, their actual significance in terms of predicting adverse outcomes such as sudden cardiac death is not completely clear. Although some studies suggest a potential connection between certain arrhythmias and cardiac events [17,30], further research is needed to elucidate the role of arrhythmias in the prognosis and management of feline HCM.

The most frequent arrythmias detected on a routine ECG that have been encountered in cats with structural heart disease were ventricular premature complexes, ranging from an incidence of 62% to 85.9% of all arrythmias, depending on different studies [37,53]. In two separate 24 h monitoring studies conducted in healthy cats, ventricular premature complexes (VPCs) were also identified as the most prevalent arrhythmia, with at least one VPC being identified in 78–90% of cats [32,36,37].

Therapy for subclinical HCM is controversial, as there is not enough evidence that certain therapies improve outcomes. Few data are available regarding anti-arrhythmic therapy for rhythm disturbances that are associated with HCM and other cardiomyopathies. Further research is needed to clarify the indications and benefits of this therapy.

In human HCM, the average heart rate and HRV can be used as indicators for prognosis [28,44]. While some studies on cat HCM did not find a difference in the average heart rate between affected cats and healthy controls [21,22,31], one study found that cats with HCM had higher mean heart rates [12]. A decrease in heart rate has been observed after a few hours of wearing the device, indicating that a period of accommodation is needed for the reduction in stress. As increased sympathetic tone could potentially cause ventricular arrhythmias [12], further research is needed to elucidate whether the stress of wearing the device could trigger arrythmias and influence the results of Holter monitoring. As a single study of HRV has been conducted on cats with HCM, no conclusion can be drawn regarding the value of HRV as a prognostic indicator. There is a considerable knowledge gap regarding HRV between human and feline medicine, so further studies are needed to clarify the association of HRV and outcomes of HCM in cats.

Different types of rhythm disturbances can sometimes be associated with extracardiac causes. Second-degree AVB, single ventricular ectopic beats, and ventricular tachycardia were observed in a cat with hyperthyroidism and no signs of cardiac disease [54]. Other possible extracardiac causes of ventricular arrhythmia can be severe anemia, electrolyte imbalances, and hypovolemia [55].

To our knowledge, there are no studies that evaluate whether the duration of Holter monitoring influences its sensitivity of detecting rhythm disturbances in cats with HCM. A study conducted on dogs showed that a 48 h Holter monitoring period does not seem to offer any significant advantages compared to a 24 h period, increasing the chance of detecting a relevant event by only 14.5% [56]. However, it has been demonstrated that a 7-day period of monitoring increases the likelihood of detecting ventricular arrythmias in Dobermann Pinchers with DCM, compared with the first 24 h of recording [57]. Further research is needed to evaluate whether the period of monitoring influences the probability of detecting relevant arrythmias in cats with HCM and other cardiomyopathies.

## 9. Conclusions

In conclusion, this review highlights the diagnostic and prognostic potential of Holter monitoring in feline hypertrophic cardiomyopathy. While ECG remains a conventional diagnostic tool, Holter monitoring offers a more comprehensive assessment of cardiac rhythm over an extended period, potentially revealing abnormalities missed by standard ECG. However, the routine use of Holter monitoring in cats with HCM is not yet established and further research is needed to define its utility more precisely.

## 10. Future Directions

Despite several studies being conducted in the last decades focusing on arrhythmias associated with HCM in cats, a significant lack of data persists. Further research is needed to evaluate the diagnostic and prognostic utility of electrocardiography and Holter monitoring, as well as the need for anti-arrhythmic therapy for certain rhythm disturbances. Establishing a standardized technique for Holter monitoring in cats could improve the quality of the examination and facilitate the consistency and reproducibility of Holter examinations.

## Figures and Tables

**Table 1 animals-14-02165-t001:** Arrythmias identified in previous studies.

Author	No. of Study Patients	Mean Age (Years)	Ventricular Ectopy						Supraventricular Ectopy			Additional Events
			No. of Cats	Median	Range	Bigeminy	Couplets	Vtach	No. of Cats	Range	SVT	
Walker	16 cats with HCM	3.43 ± 1.22	16/16	10	1–202	NM	1 ^a^	4/16 ^a^	9/16	1.5 (0–200) ^b^	1 ^a^	
	7 control cats	4.13 ± 1.51	6/7	6	0–151	NM	0 ^a^	2/7 ^a^	6/7	2 (0–23) ^b^	0 ^a^	
Bartoszuk	16 cats with cHCM	9.5 (1.5–16.5)	15/16	431	0–18,919	NM	NM	8/16 ^a^ 11.5 (1–91) ^b^	2/16	357; 1603 ^c^	2/16 ^a^ (1; 1) ^c^	Complete AVB (*n =* 1) ^a^
	15 cats with dHCM	6 (1–14.5)	15/15	867	1–35,160	NM	NM	6/15 ^a^ 2 (1–1497) ^b^	2/15	12,498; 28,568 ^c^	2/15 ^a^ (135; 1075) ^c^	AF (*n =* 2) ^a^
	10 control cats	4.25 (1–7.5)	8/10	2	0–13	NM	NM	0	0	-	0	
Jackson	17 cats with HCM	7.8 ± 3.5	17/17	124 ^d^		NM	NM	NM	15/17	9 ^d^	NM	
	15 control cats	6.0 ± 2.8	14/15	4 ^d^		NM	NM	NM	9/15	1 ^d^	NM	
Hanås	15 cats with HCM	4.5 (1.4–11.2)	12/15	3	0–1745	5 ^a^	4 ^a^	NM	3/15	2; 2; 4 ^c^	NM	
Ferasin	7 RCM, 3 ARVC, 3 UCM	6.9 ± 4.7	13/13	2031	338–8305	6/13 ^a^	11/13 ^a^	4/13 ^a^	7/13	1–74	NM	Complete AVB, PAS, AF
Yamaki	20 healthy cats	NM	6/20	NM	1–18	NM	NM	NM	2/20	3; 4 ^c^	NM	
Hanås	23 healthy cats	5.5 (1–15)	18/23	3	0–146	4 ^a^	3 ^a^	0	1/23	1 ^c^	0	
Ware	10 healthy cats	1–4	8/10	3.8 ± 0.9		NM	2 ^a^	0 ^a^	2/10	1; 1 ^c^	NM	
	10 healthy cats	8–14	10/10	96 ± 77.6		NM	4 ^a^	1 ^a^	6/10	1–21	NM	

^a^—number of affected cats; ^b^—median and range for affected cats; ^c^—number of events in individual cats with the abnormality; ^d^—geometric mean, NM—not mentioned; AVB—atrioventricular block, PAS—persistent atrial standstill, AF—atrial fibrillation, RCM—restrictive cardiomyopathy, ARVC—arrhythmogenic right ventricular cardiomyopathy, UCM—unclassified cardiomyopathy.

**Table 2 animals-14-02165-t002:** Holter monitoring technique and environment of monitoring.

Author	Year	No. of Electrodes	Channels	Placement	Environment
Walker	2022	5	3	3 on the right, 2 on the left	Research laboratory
Ferasin	2020	5	NM	2 on the right, 3 on the left	Hospital
Bartoszuk	2019	3	NM	2 on the left, 1 on the right	Hospital
Hanås	2015	NM	NM	NM	Home
Jackson	2014	5	2	NM	Home
Yamaki	2014	NM	3	NM	Home
Hanås	2009	7	3	NM	Home
Ware	1999	5	2	3 on the right, 2 on the left	Home

NM—not mentioned.

## Data Availability

No new data were created or analyzed in this study. Data sharing is not applicable to this article.

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
