# Peer review of "The Unseen Side of Feline Hypertrophic Cardiomyopathy: Diagnostic and Prognostic Utility of Electrocardiography and Holter Monitoring"

_animals, 2024, doi:10.3390/ani14152165_

Round 1

Reviewer 1 Report

Comments and Suggestions for Authors

Dear authors,

The authors presented the paper "The unseen side of feline hypertrophic cardiomyopathy: diagnostic and prognostic utility of electrocardiography and Holter 3 monitoring", which aimed to summarise the knowledge of ECG AND Holter AND their utility in diagnosing HCM in cats. In my opinion, the article could have been at least twice as short if the principles of narrative review had not been followed rather than describing each study separately. Furthermore, the intricacy of some passages makes it difficult to read. 

I would also like to draw attention to the low level of presentation of the article. There is a lack of electrocardiograms, scans from sympathetic balance analysis or printouts from Holter analysis.

Inaccuracies in the text:

Line 41 - please distinguish between races, as the incidence is different between races and hybrids. 

Line 44 - please be more specific about the causes that may cause hypertrophy and mimic HCM (aortic stenosis)

Line 45 - the authors probably mean that despite wall thickening they remain asymptomatic or the hypertrophy itself is not severe. Please explain this view on a genetic basis. Differences between homozygotes and heterozygotes with respect to a given allele.

Line 59 - please do not forget to mention micro-CT and MRI in the diagnostic methods, these are methods that especially in recent times are strongly developed.

Line 69 - please provide a citation.

Line 95 - please rephrase this paragraph "2. Ventricular and supraventricular arrythmias on Holter monitoring" Please reframe your argument, as dozens of lines of repetitive opposites and verbal repetitions make this text difficult to understand. Please make the statement more complex so that the reader can understand the authors' main point about the usefulness of ECG and Holter in the assessment/diagnosis of HCM. This is missing so far. Furthermore, I would like to divide the paragraph into VPC and APC.

114 - this wording is not needed. Rather, in the case of data that indicate a trend and are not significant, one determines whether they were significant or not.

122 - please remove the wording about the number of control group. Throughout the text where it is not necessary, please remove the indication of the number of control group. Such data can be included in the table.

211 - seemed or was there a lower heart rate?

215 - too many mental shortcuts are used by the authors. First they talk about several studies, of which they then distinguish three, and again they talk about some one. And we learn about the fact that it took place in a hospital environment in a separate sentence. Moreover, citation 6 is not included in these 'some studies.....[10,11,22].'

 This way of writing unfortunately makes the text difficult to understand. The reader may get lost as in subsection no. 2.

"Some studies that used Holter monitoring demonstrated that there is no difference 214 in mean heart rate between HCM affected cats and healthy cats [10,11,22]. In two of these 215 studies, Holter monitoring was conducted at home [10,22], and in the third one, in the 216 hospital environment [11]. However, one study showed that cats with cHCM and dHCM 217 had higher mean heart rates than healthy controls, but there was no difference between the 218 two HCM groups. This study was conducted in a hospital environment [6]."

233 - profiles appear to differ, do they differ? Did they reach staystance significance? Or is it only possible to identify a trend based on the results?

237 - Holter monitoring technique - this paragraph does not fit here at all. This was supposed to be an analysis of the usability of Holter and ECG. Please move this paragraph to the appropriate place or delete it.

292 - you have not demonstrated this, it was a known problem already. Alternatively it has been summarised.  

Best regards

Reviewer 2 Report

Comments and Suggestions for Authors

The review by Cofaru et al. summarizes current data concerning HCM in cats, and different diagnostic tools. I found this paper quite interesting and clearly written. I have only several suggestions on what kind of information can be included and/or strengthened.

1. T-wave inversion as an ECG sign of HCM and a potential predictor of HCM outcomes.

2. Mechanistic relationships between HCM and arrhythmias (not necessarily in cats).

3. I am not qualified to evaluate English, but it seems to me that I saw several linguistic defects. Could you double check the text?

Reviewer 3 Report

Comments and Suggestions for Authors

Authors present a review related to diagnostic and prognostic utility of electrocardiography and Holter monitoring in cats with HCM. The review is well written and comprehensive, however, I have some suggestions which I will list below.

Line 161 – this phrase is difficult to understand and should be rephrased.

Line 166 – it should be stated here that the investigation was performed in healthy cats only

Line 170 – what do authors mean by arrhythmia by impulse formation. Can authors explain better for the reader? Thank you.

Line 202 – I would not call “the characteristics of LV enlargement” a rhythm disturbance. There are ECG morphological changes. Moreover, in this paragraph it would be useful to add information regarding the ECG recording duration in these studies for the reader.

Line 226: what do authors mean by “The gold standard measurement for heart rate variability is the standard deviation of the inter beat interval of normal sinus beats (SDNN)”? SDNN is one way to express the degree of HRV using the time domain analysis. However, there are multiple parameters that may be assessed such as PNN50% or rMSSD. Moreover, frequency domain analysis expressed the different variations in frequency expressed as HF, LF or LF/HF. There are also non-linear methods of R-R interval variation analysis. Therefore, I suggest rephrasing this using an expression such as: “One of the measures of HRV is SDNN…. “.

Line 229: change “if” to “in”

Heart rate variability paragraph: in this reviewer's opinion HRV should not be included in this review. Authors describe the finding in only one study on cats with HCM. I do not think this is enough to make any comments and especially since there is not enough data to compare the results of more than one study, or to draw any strong conclusions. However, if authors chose not to remove the HRV data in this review, I suggest to add at least the available data in healthy cats and only then introduce the HRV data of HCM cats.

Round 2

Reviewer 1 Report

Comments and Suggestions for Authors

Thank you for all the amendments. Please refer to the following. 

"Commnets 2"

Breeds such as the Maine Coon I Randoll are predisposed to HCM-related genes. A homozygous allele pattern in terms of MYBP3 mutations in Maine Coon and Ragdoll lead to the development of HCM, it is indisputably established. Recent studies have even pointed to this conclusion. Please explore the literature on the genetic basis of HCM in cats and correct this passage.

Best Regards

Author Response

Thank you for your feed-back and we are very pleased that the improvements we made based on your recommendations enhanced the quality  of our paper.

Comments 2: „Breeds such as the Maine Coon I Randoll are predisposed to HCM-related genes. A homozygous allele pattern in terms of MYBP3 mutations in Maine Coon and Ragdoll lead to the development of HCM, it is indisputably established. Recent studies have even pointed to this conclusion. Please explore the literature on the genetic basis of HCM in cats and correct this passage.”

Response 2: Thank you for this observation! We took into consideration this suggestion and therefore we corrected this passage based on the most recent studies.